# Global Distribution of *Aedes aegypti* and *Aedes albopictus* in a Climate Change Scenario of Regional Rivalry

**DOI:** 10.3390/insects14010049

**Published:** 2023-01-03

**Authors:** Gabriel Z. Laporta, Alexander M. Potter, Janeide F. A. Oliveira, Brian P. Bourke, David B. Pecor, Yvonne-Marie Linton

**Affiliations:** 1Graduate Research and Innovation Program, Centro Universitario FMABC, Santo André 09060-870, SP, Brazil; 2One Health Branch, Walter Reed Army Institute of Research, Silver Spring, MD 20910, USA; 3Walter Reed Biosystematics Unit, Smithsonian Museum Support Center, Suitland, MD 20746, USA; 4Department of Entomology, Smithsonian Institution—National Museum of Natural History (NMNH), Washington, DC 20560, USA; 5Department of Civil Engineering, School of Engineering, Campus Crajubar, Universidade Regional do Cariri, Crato 63105-010, CE, Brazil

**Keywords:** *Aedes*, climate change, climate models, environmental indicators, forecasting, statistical model, risk factors

## Abstract

**Simple Summary:**

*Aedes aegypti* and *Aedes albopictus* mosquitos pose threats of arboviral disease emergence to humans in future climates. Unique mosquito georeferenced data from VectorMap and comprehensive environmental data from WorldClim v. 2.1 were herein used to analyze a global scenario of climate change under a socioeconomic pathway of regional rivalry. Results showed that this shared socioeconomic pathway is likely to affect vector-borne diseases. Climate changes on both vector species distributions will surely have major impacts on public health decision making.

**Abstract:**

Arboviral mosquito vectors are key targets for the surveillance and control of vector-borne diseases worldwide. In recent years, changes to the global distributions of these species have been a major research focus, aimed at predicting outbreaks of arboviral diseases. In this study, we analyzed a global scenario of climate change under regional rivalry to predict changes to these species’ distributions over the next century. Using occurrence data from VectorMap and environmental variables (temperature and precipitation) from WorldClim v. 2.1, we first built fundamental niche models for both species with the boosted regression tree modelling approach. A scenario of climate change on their fundamental niche was then analyzed. The shared socioeconomic pathway scenario 3 (regional rivalry) and the global climate model Geophysical Fluid Dynamics Laboratory Earth System Model v. 4.1 (GFDL-ESM4.1; gfdl.noaa.gov) were utilized for all analyses, in the following time periods: 2021–2040, 2041–2060, 2061–2080, and 2081–2100. Outcomes from these analyses showed that future climate change will affect *Ae. aegypti* and *Ae. albopictus* distributions in different ways across the globe. The Northern Hemisphere will have extended *Ae. aegypti* and *Ae. albopictus* distributions in future climate change scenarios, whereas the Southern Hemisphere will have the opposite outcomes. Europe will become more suitable for both species and their related vector-borne diseases. Loss of suitability in the Brazilian Amazon region further indicated that this tropical rainforest biome will have lower levels of precipitation to support these species in the future. Our models provide possible future scenarios to help identify locations for resource allocation and surveillance efforts before a significant threat to human health emerges.

## 1. Introduction

*Aedes aegypti* (Linnaeus, 1762) and *Ae. albopictus* (Skuse, 1895) are two of the most prolific mosquito vector species in the world. Both species are found on every continent—except the Antarctic—and have played major roles in various vector-borne disease outbreaks over the last century [1,2,3]. They are known vectors of the viruses that cause yellow fever (YFV), dengue (DENV), chikungunya (CHIKV) and Zika (ZIKV) [4,5,6,7,8]. The distribution of these species is also changing, driven in part by global climate change [9]. Distribution modelling studies have demonstrated that the distribution of these species are expected to dramatically change during the next century as habitats that have previously been inhospitable become favorable [10,11]. These changes in distribution patterns will represent significant challenges to public health, as mosquito surveillance and control strategies will not be able to rely on historical population trends. In short, the predicted magnitude and pace of climate change will render our historical experience obsolete [12]. Both species have successfully spread across the globe in part because of their close association with humans, but also because the eggs of both species can become desiccated for long periods of time and still be viable once exposed to water [13]. Resistance to desiccation facilitates passive transportation of eggs through global trade, thereby triggering *Aedes* invasion into new locations [14]. One key example of this passive dispersal is the repeated migration events led by the used-tire shipping industry [14].

VectorMap—a product of the Walter Reed Biosystematics Unit (WRBU)—is the world’s largest and most comprehensive web-based repository for arthropod vector collection data [15]. The VectorMap project has captured vector surveillance data for all key vector taxa, but the greatest number of records belong to *Ae. albopictus* and *Ae. aegypti*. The dataset includes the most significant publication of surveillance data for these two species in recent years [2]. In addition to data extracted from the literature, VectorMap also contains numerous specimen records from the United States National Museum (USNM) collection and numerous WRBU collaborators who have submitted their data to be included in the VectorMap database.

Currently, *Ae. aegypti* is established in 167 countries [16]. *Aedes aegypti* is human-centered, as it shows multiple events of synanthropic adaptation worldwide [17,18]. It is this close association with humans that has made this species such an important vector of pathogens to humans. *Aedes aegypti* environmental suitability has expanded globally by as much as 1.5% per decade since 1950 and this trend is expected to accelerate over the next century [9]. In Latin America and the Caribbean, extreme events of above-average temperatures are causing *Ae. aegypti* populations to adapt to new habitats including underground oviposition and resting sites [19]. Even areas that are expected to receive far less rainfall may be at risk [20]. In East Africa, increased use of water storage containers by humans in response to drought conditions has led to a rise in CHIKV cases as *Ae. aegypti* will readily lay eggs, and immatures can raise, in these containers [20].

Originally native to southeast Asia, *Ae. albopictus* is now known to occur in 126 countries world-wide [16]. It’s range is also expected to expand as temperatures rise [21]. Both winter and summer temperatures are the most significant factors currently limiting *Ae. albopictus* distribution in southern Europe and preventing its expansion to northern and western Europe [21]. *Aedes albopictus* will readily exploit artificial containers for immature development, and can exist in sympatry with *Ae. aegypti* immatures. This species will also utilize a wide variety of natural environments which may give them an edge when competing against other species. There is also evidence that *Ae. albopictus* is much more resilient than *Ae. aegypti* in areas that experience frequent temperature changes, such as more temperate areas [22]. This flexibility may allow *Ae. albopictus* to outcompete *Ae. aegypti* in certain areas, further complicating our knowledge and predictions of species distributions.

Several species distribution models (SDMs) have been developed for these important mosquito species in recent years [1,9,23,24,25,26]. However, previous models are limited to specific geographic regions and incorporate a limited number of surveillance records. Here, we present SDMs predicting the global distribution of *Ae. aegypti* and *Ae. albopictus,* developed using the most comprehensive database of occurrences ever assembled. We have developed global SDMs using environmental layers (temperature, precipitation), future climate projections through to the year 2100 in twenty-year spans, and a database of occurrence data.

It is our main goal to present global distributions of *Ae. aegypti* and *Ae. albopictus* in a climate change scenario of regional rivalry. The Intergovernmental Panel on Climate Change (IPCC) proposed five Shared Socioeconomic Pathway (SSP) scenarios in the 2021 report [27]. The first is SSP1 (+1.9 W/m^2^), that represents the most optimistic scenario because it assumes a global warming below 1.5 °C by 2100 and a lower frequency of extreme events in the century [27]. At the opposite end, SSP5 (+8.5 W/m^2^) is the most pessimistic, assuming the highest warming of future greenhouse gas emissions [27]. At intermediate positions, SSP2-4.5 and SSP1-2.6 are scenarios with stronger mitigation of greenhouse gas emissions and thus lower future warming levels [27]. Lastly, SSP3-7.0 is the regional rivalry scenario and is characterized by increasing levels of CO_2_ emissions that are assumed to double by 2100 [27]. More importantly, the SSP3-7.0 scenario assumes that mitigation of greenhouse gas emissions and adaptation for socio-economic challenges will be undermined by regional rivalry [28]. Evidence of regional rivalry has been seen with the reemergence of right-wing extremism in Brazil and elsewhere (US and Europe), which raises uneasiness about resolving issues of global concern, including Amazonian conservation, pandemics and climate change [29].

## 2. Materials and Methods

### 2.1. Data Description

Occurrence data for *Ae. aegypti* and *Ae. albopictus* were downloaded from VectorMap (vectormap.si.edu), developed and maintained by the Walter Reed Biosystematics Unit (WRBU). Data were compiled from three sources: (1) peer-reviewed literature, (2) museum specimen records, and (3) publicly available mosquito surveillance records. VectorMap documents 93 fields of information for each record, including taxonomy, location site description, and collection method. Collection sites are georeferenced using the point radius method using coordinates reported by authors/data submitters and named places were assigned coordinates from an online gazetteer [30,31]. A search of the VectorMap mosquito map service was carried out on January 12th, 2022. The use of Species = “aegypti” returned 32,251 total records, while Species = “albopictus” returned 43,987 total records. Literature offered the greatest number of presence records with data generated from 29 articles. Additional records were sourced from USNM mosquito specimen records digitized and uploaded to VectorMap by WRBU. Finally, data were also sourced from surveillance records submitted to VectorMap by the Armed Forces Health Surveillance Division-Global Emerging Infections Surveillance (AFHSD-GEIS) global network.

### 2.2. Data Preparation

The crude data were imported into the R v. 4.0.4 program (The R Foundation for Statistical Computing, Vienna, Austria) using tidyverse v. 1.3.1 (tidyverse.org) by selecting three columns (‘Species’, ‘Decimal Latitude’, and ‘Decimal Longitude’) and by filtering the species type (*Ae. aegypti* or *Ae. albopictus*) to generate a species occurrence data per each species.

Two procedures for the preparation of data were further applied on the species occurrence data. The first procedure removed all duplicates, i.e., specimens captured in the same location, to guarantee that each point coordinate had a unique value of latitude and longitude. This procedure was undertaken by using the R function duplicated in the base package. A map of world country administrative boundaries with oceans in the maptools package was overlapped by the remaining point coordinates. All point coordinates were checked and any landing in the ocean were removed. The final species occurrence data for *Ae. aegypti* (*N* = 9735) and *Ae. albopictus* (*N* = 13,093) were free of duplicated and non-land georeferenced latitudes and longitudes.

### 2.3. Environmental Data

Global climate data for 1970–2000 and future climate data for 2021–2100 were obtained from WorldClim v. 2.1 (worldclim.org). Thirty-six environmental layers with spatial resolution of ~4 × 4 km^2^ were selected, as follows: (layers 1–12) average minimum temperature (°C) January–December; (layers 13–24) average maximum temperature (°C) January–December and (layers 25–36) total precipitation (mm) January–December. These variables were selected because of their strong link with the ecological niche of both species, thereby having a combined relative contribution as high as 77.6% and 74.3% among all covariates on *Ae. aegypti* and *Ae. albopictus* global distribution, respectively [1]. They were employed to build a fundamental niche model and then estimate future potential distributions.

The SSP3 (+7.0 W/m^2^) was the scenario chosen for the projections of future climate data [28]. This was selected because it represents a likely scenario for this century. The emergence of right-wing extremism in Brazil and in other liberal countries (e.g., USA) can yield regional rivalry. Regional conflicts, including the current war in Ukraine, will push global issues (e.g., control of greenhouse gas emissions) into the background and tropical rainforests will be disproportionately impacted. This scenario is already occurring in the Brazilian Amazon [29].

The Global Climate Model (GCM) herein selected in WorldClim v. 2.1 had been downscaled from the Coupled Model Intercomparison Project phase 6 (CMIP6). The GCM selected here was the Geophysical Fluid Dynamics Laboratory Earth System Model v. 4.1 (GFDL-ESM4.1; gfdl.noaa.gov) of the National Oceanic and Atmospheric Administration (NOAA). This 4th generation of an earth system model contains future outcomes of temperature and precipitation based on interactions of major components of natural and anthropogenic processes (CO_2_, dust, iron, and nitrogen) that influence the climate system.

Future time periods considered for the modelling of potential distributions of *Ae. aegypti* and *Ae. albopictus* were 2021–2040, 2041–2060, 2061–2080, and 2081–2100.

### 2.4. Model Building, Fitting, and Evaluation

Species occurrence data (i.e., presence data) were randomly sorted into training data (75% of presence data) and testing data (25% of presence data). Training data were utilized for model fitting, whereas testing data were used for model evaluation.

Pseudo-absence data were obtained by selecting random point coordinates of training (*n* = 250) and testing (*n* = 62) from probability of occurrence = 0 (blue color) of the global maps of predicted distribution of *Ae. aegypti* and *Ae. albopictus* [1]. This approach is used for generating the non-presence class for generalized boosted regression models [32]. It means unsuitable places where species absences are more likely, a.k.a. pseudo-absence [33].

A generalized boosted regression tree model fitting was run using the R function gbm.step in the dismo package. The dataset of analysis consisted of values of presence(=1)/pseudo-absence(=0) of each species and geographically corresponding extracted values of environmental layers. This geographical correspondence was attained by extracting values of environmental layers per point coordinates of presences and pseudo-absences. The dataset of analysis (*y* = presence/pseudo-absence, *X*_1–36_ = environmental layers) was divided into 10 subsets of equal stratification between presences and pseudo-absences for fitting 10 submodels that had their results cross-validated according to [34]. Deviance from results of cross-validation among the submodels was calculated step-by-step. In each step, forward model complexity (number of regression trees) increased to improve model fitting and thus decrease deviance. The optimum complexity (i.e., the minimum deviance) was selected to adjust data and estimate contributions of environmental variables on presence/pseudo-absence of each species.

The fitted model with the training data was then evaluated by assessing its accuracy level to correctly predict presences and pseudo-absences in the testing data. The number of errors of omission (missing presences) and commission (missing pseudo-absences) vs. the number of correctly identified presences and pseudo-absences were used to calculate a cut-off at which the sum of sensitivity (true positive rate) and specificity (true negative rate) is highest.

### 2.5. Spatial Prediction

The fitted and evaluated model was applied to spatially predict the probability of presence of *Ae. aegypti* and *Ae. albopictus* across all continents in the world. The probability of presence meant a fundamental niche to the global climate data for 1970–2000 and meant potential distribution to each future climate data period (2021–2040, 2041–2060, 2061–2080, and 2081–2100). Technically, a raster object was built with predictions from the fitted model over environmental raster data using the R function predict in the raster package. This approach has been commonly used for species distribution modelling elsewhere [32].

## 3. Results

### 3.1. Species Occurrence Data

*Aedes aegypti* occurrence data showed 9,735 presence records in 114 countries (Figure 1A). Country distribution was unequal towards countries in the tropical and subtropical zones, including Brazil (*n* = 4780; 49.1%), Taiwan (*n* = 817; 8.4%), Indonesia (*n* = 421; 4.3%), Thailand (*n* = 398; 4.1%), United States (*n* = 337; 3.5%), India (*n* = 331; 3.4%), Mexico (*n* = 328; 3.4%), Kenya (*n* = 240; 2.5%), Australia (*n* = 169; 1.7%), and Viet Nam (*n* = 156; 1.6%) with highest numbers of presence records among the 114 countries (Figure 1A).

*Aedes albopictus* presence records (*n* = 13,093) were noted from 85 countries (Figure 1B). This occurrence data showed distribution towards higher latitudes in the northern hemisphere (Europe, United States, and Japan) in addition to the expected tropical and subtropical countries (e.g., Brazil). United States of America (*n* = 6135) and Brazil (*n* = 3360) contributed 72.5% of occurrence data (Figure 1B).

### 3.2. Environmental Data

Climate data 1970–2000 and future climate projections 2021–2100 of GFDL-ESM4.1 in SSP3-7.0 showed variation of precipitation and temperature in northern and southern hemispheres. Temperature showed a linear increase over time in both hemispheres. Precipitation increased or decreased over time depending on the region of the world. In Washington DC, the capital of the United States, annual average temperature and annual total precipitation increased from 1100 mm and 13.2 °C in 1970–2000 to 1269 mm and 17.1 °C in 2081–2100. In Brasilia, the capital of Brazil, the annual average temperature also increased by 3.9 °C (range: 20.3–24.2 °C), but annual total precipitation decreased by 207 mm (range: 1539–1332 mm) in 2081–2100 vs. 1970–2000. Future climate projections showed that both northern and southern hemispheres will become hotter in the future, but the northern hemisphere will get wetter, whereas the southern hemisphere will get drier.

### 3.3. Boosted Regression Tree Model

The fitted model for *Ae. aegypti* had 1450 iterations performed. The evaluation showed that the fitted model had very good reliability and was able to classify presence and pseudo-absence correctly (Appendix A). The receiver operating characteristic (ROC) area under the curve (AUC) was 0.995 and the threshold that maximized the sum of sensitivity and specificity was 0.845. This threshold was further used in the spatial prediction (see below in Section 3.4). The relative influences of environmental variables on the fitted model were 50.71% of average maximum temperature, 24.02% of average minimum temperature, and 25.27% of total precipitation. The most important predictor was maximum temperature in October (relative contribution = 15.4%). Relative contribution and response curves per top 15 predictors are shown in Appendix A. A more precise assessment at fitted values in relation to the most important predictor in the model (maximum temperature in October) is in Appendix A. The optimum range of average maximum temperature was between 18–37 °C. The lowest average minimum temperature should exceed 10 °C for higher probabilities of presence. Monthly precipitation above 10 mm and up to 280 mm had the maximum probability of occurrence.

The fitted model for *Ae. albopictus* had less iterations performed than the one for *Ae. aegypti*, equaling 1050. Evaluation of this model showed that its AUC was approximately 1.0 and its threshold that maximizes sensitivity + specificity was 0.995. This fitted model was thus considered highly reliable in predicting the probability of presence of *Ae. albopictus* (Appendix A). Additionally, the relative influences of environmental variables on this fitted model were different when compared to those of the previous fitted model for *Ae. aegypti*. For the fitted model of *Ae. albopictus*, the relative contribution was 49.48% of average minimum temperature, 32.62% of total precipitation, and 17.91% of average maximum temperature. The top three predictors were: minimum temperature in October (relative contribution = 22.6%), precipitation in April (20%), and minimum temperature in September (17.8%) (Appendix A). This species can tolerate average minimum temperatures down to 6 °C. The optimum range of average maximum temperature 9–31 °C was cooler than that of *Ae. aegypti*. The optimum range of total precipitation was narrower in *Ae. albopictus* and was estimated at between 50–230 mm monthly.

### 3.4. Fundamental Niche and Potential Future Distributions

The threshold values obtained from the evaluation step of the fitted models were further applied to transform the spatially predicted probability of presence to a binary score that represented two outcomes: (1) fundamental niche (predicted probability of presence higher than the threshold), (2) unsuitable habitats (≤threshold value).

Fundamental niche in Figure 2A,B meant regions having suitable climate for these species where they could be present if they are able to disperse into and find favorable biotic interactions. Antarctica, Canada, Russia, Scandinavia, Sahara Desert, southern Argentina, the Australian deserts, and the New Guinea Highlands were all determined unsuitable habitats for both *Ae. aegypti* and *Ae. albopictus* (Figure 2A,B).

The fundamental niche of *Ae. aegypti* showed higher association with regions of the southern hemisphere in South America, Sub-Saharan Africa, Indonesia, and northeastern Australia (Figure 2A). Additionally, suitable regions in Mexico, United States, North Africa, and Southeast Asia were also present (Figure 2A). Europe showed no suitability, except for scattered coastal areas in Portugal, Spain, and Italy (Figure 2A).

In contrast to the fundamental niche of *Ae. aegypti*, the fundamental niche of *Ae. albopictus* shown in Figure 2B is less associated with southern regions but considerably associated with the northern hemisphere, including regions of the United States, Europe, China, and Japan (Figure 2B).

Potential future distributions showed important changes over the four periods (2021–2040, 2041–2060, 2061–2080, 2081–2100) depending on species, hemisphere, and continent being analyzed.

*Aedes aegypti* potential future distributions in Americas showed that the United States will not develop additional suitable areas, while most of the Amazon basin of Brazil will become unsuitable for this species in the future (Figure 3A–D).

Europe showed suitable areas in Portugal, Spain, France, and Italy, while parts of Sub-Saharan Africa would have fewer suitable areas (Figure 4A–D).

Some areas of China, Japan, and southern Australia showed suitability for future *Ae. aegypti* (Figure 5A–D).

Considering that *Ae. albopictus* was shown to be more associated with the northern hemisphere (Figure 2B) and have higher tolerance to cooler areas than *Ae. aegypti* (see Section 3.3 above), future climate change scenarios showed that this species’ potential distribution will likely expand in the United States (Figure 6A–D). However, South America would possess fewer suitable areas, particularly in the Amazon basin (Figure 6A–D).

*Aedes albopictus* potential distribution will likely expand towards northern areas in Europe (Figure 7A–D). On the contrary, Sub-Saharan Africa will possess less suitable habitats in the future (Figure 7A–D).

Indonesia also showed that future suitability for *Ae. albopictus* will be scarcer in the future southern hemisphere but will remain somewhat unchanged in Asia (Figure 8A–D).

In addition to the maps of binary outcomes (fundamental niche, unsuitable habitats) shown in Figure 2, Figure 3, Figure 4, Figure 5, Figure 6, Figure 7 and Figure 8, maps of probability of presence are shown in Appendix A.

## 4. Discussion

Our results indicate that average maximum temperatures of 10–30 °C and monthly precipitation of 50–200 mm were the most suitable ranges for both species. These variables had a joint contribution of 75% for the prediction of fundamental niches in a previous modelling study [1]. This explains the potential distributions of these species in areas of tropical and subtropical zones, where they can transmit arboviral pathogens to humans in high density cities [35]. This further points to their niche similarities and overlapping co-occurrences [26]. However, differences between *Ae. aegypti* and *Ae. albopictus* were seen here when comparing their spatial predictions from fitted boosted regression tree models. *Aedes aegypti* showed a generally broader distribution potential globally. An important exception is the potential of *Ae. albopictus* distribution across temperate zones. This *Ae. albopictus* potential has been studied previously in other modelling studies [11,23,24,25,26]. Its northern range expansion in the United States and southern Canada was predicted by the end of this century [11,23,24,26]. Future models shown here and those published elsewhere [26] also agree that further expansion of *Ae. albopictus* towards northern Europe is likely in 2050. In the Southern Hemisphere, its distribution is expected not to expand but likely shrink in the future.

*Aedes aegypti* has a markedly broad distribution globally, as shown here and elsewhere [9,10,36]. Its potential distribution is predicted to increase 10–30% globally, depending on the scenario of greenhouse gas emissions (most optimistic–pessimistic) by the end of this century [36]. Other studies predicted that this trend will accelerate after 2050, which means that its expansion may be greater than 30% in comparison with the current distribution [9]. This is because more suitable habitats for *Ae. aegypti* in many areas in the Northern Hemisphere are expected over the next century [10]. Conversely, here and elsewhere [10] showed that favorable suitable habitats across many regions in the Southern Hemisphere are expected to recede as the impacts of climate change increase.

More specifically, we observed that the today’s climatically favorable habitats for both *Ae. aegypti* and *Ae. albopictus* are predicted to be contracted across the Amazon under the future scenario of regional rivalry herein chosen. Amazonian rainforest heavily depends on suitable climate, including frequently high precipitation levels, meaning that there is a high likelihood of largescale diebacks of plant species due to intensifying dry seasons [37,38]. This loss of primary tropical rainforest due to increasingly frequent droughts in the Amazon region will also accelerate the continued decline of biodiversity [39]. Niche displacement of *Nyssorhynchus darlingi* (formerly known as *Anopheles darlingi* [40]), the dominant malaria vector in the region, has already been shown under climate change scenarios [32]. Considering how successful these species have been at colonizing the world’s tropical zones, it is worrisome that places including the Amazon rainforest may become inhospitable to them due to the loss of optimum precipitation levels over the next century.

Europe will continue providing suitable habitats for both species in this century. Arguably, we have already seen a preview of this potential impact with the 2012 outbreak of dengue fever on the island of Madeira, Portugal. Prior to 2005, *Ae. aegypti* had not been known to occur on the island, but at least two introduction events (originating from Venezuela) established a population on the island [41]. In 2012, island residents experienced their first cases of autochthonous dengue infection, spread by *Ae. aegypti* mosquitoes [42]. The expansion of *Ae. albopictus* further into the European continent is a public health issue of current concern. This species is contributing to an increased risk of vector-borne disease transmission across mainland France [43]. A total of 65 autochthonous dengue cases (serotypes 1 and 3) were reported in France, 2022, and the dengue virus identified in these humans was probably vectorized by local populations of *Ae. albopictus* [43]. As many European countries are among the most visited destinations in the world [44], further invasion and transmission of serotypes of dengue or other arboviruses is likely.

The rise of political instability across the globe does not bode well for limiting the range expansion of these two species. Regional rivalries over territory and resources will likely drive human contributions to climate change for the rest of the century. In addition, robust mosquito surveillance systems must be established with data shared across borders if range expansions are to be effectively monitored. This may prove difficult during conflict, which has a long history of driving vector-borne disease incidence [45]. Regional rivalry within a country, including Brazil, will likely increase both social inequality and risks of vector-borne disease spillover [29]. Political, social, and economic instability within Venezuela has resulted in dramatic increases in vector-borne disease transmission and the resulting large-scale human migration has had considerable public health impacts for other countries in South America and as far as Europe [46,47].

We recognize there are a few notable limitations to this study. First, our model relies heavily on two environmental variables: temperature and precipitation. While we recognize that there are many other environmental factors that contribute to establishing a niche for each species, these two variables are likely the most significant factors [1,35]. The life cycle of these mosquitoes depends on the availability of habitats for eggs, larvae, and pupae [16]. The development of these immatures is only achieved when an optimum range of precipitation occurs [16]. Adult males and females depend on optimum ranges of temperature to fly into the air; otherwise, they will become inactive and/or die [16]. Second, we rely on pseudo-absence rather than true absence data. Since pseudo-absence data is derived from randomly selected locations within the study area, and occurrence data is often biased towards areas that are easier to sample, our results are biased to some extent [33]. Nevertheless, our results are supported by another study that applied a process-based mathematical model and reached similar outcomes [36]. Finally, we relied heavily on unverified identification records that were not confirmed by molecular identification of cryptic taxa. *Aedes aegypti* and *Ae. albopictus* are two of the most recognizable mosquito species and are identified quite easily by non-experts using morphology alone. Furthermore, both species are generally not considered species complexes, although there appears to be emerging evidence of cryptic structure for *Ae. aegypti* on islands in the southwestern Indian Ocean [48], and for *Ae. albopictus* in southern China and Vietnam [49].

One of the approaches that has been used to improve taxonomic resolution of SDMs is to undertake modelling only with molecularly confirmed samples, such as those identified using the Barcode of Life Data System (BOLD). This approach has been successfully applied to discriminate species in the Albitarsis Complex that can dominate malaria transmission in future scenarios [32]. It has also been used for deriving the unique niche of each cryptic species in the Strodei Subgroup of *Nyssorhynchus* [50].

BOLD uses the Refined Single Linkage (RESL) clustering algorithm to delineate operational taxonomic units called Barcode Index Numbers (BINs; [51]), which can be considered molecular-based proxies for species. BOLD BINs can therefore be considered an additional, readily accessible, and powerful dataset for the SDM analysis of cryptic species. Better curation of the BOLD database would first need to be undertaken, however, to exclude or correct specimens that have been incorrectly identified. An example of corrections needed for BOLD data can be seen with *Ae. aegypti*, where considerable specimen misidentification has currently delimited nine BINs with *Ae. aegypti* members, but where *Ae. aegypti* sensu stricto (as defined in [48]) is represented only by BIN BOLD:AAA4210. With due consideration for such curatorial issues, our recommendation is for the inclusion of BOLD BIN data in the occurrence data of the species being modelled to improve the biological accuracy of SDMs.

## 5. Conclusions

Countries all over the globe will need to prepare for vector species distributions to change and these changes will have major impacts on public health decision making. Our SDMs provide a glimpse into possible futures which can help identify locations for resource allocation and surveillance efforts before a significant threat to human health emerges. In areas where these species are expected to decline, this information can be used to justify new public health priorities.

## Figures and Tables

**Figure 1 insects-14-00049-f001:**
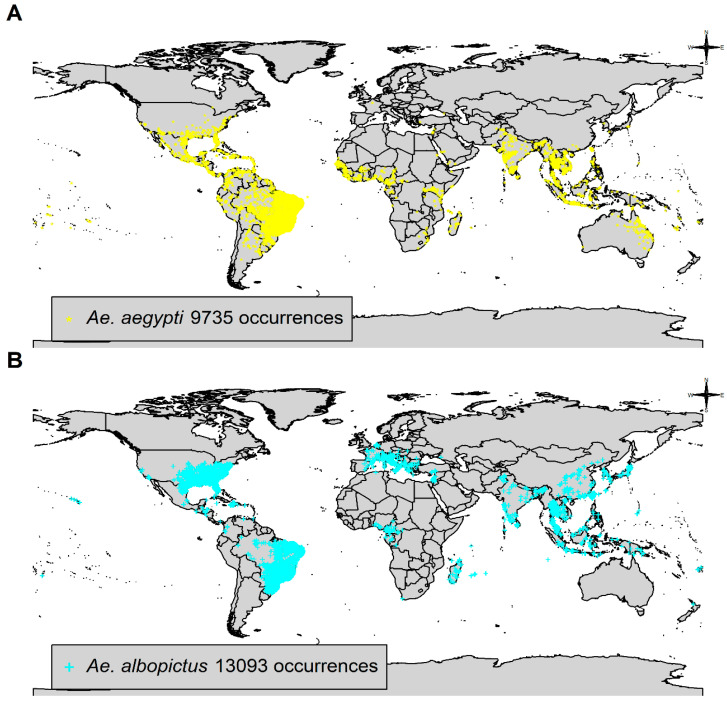
VectorMap species occurrence data: (**A**) *Aedes aegypti;* and (**B**) *Aedes albopictus* as adults (females, males) or immatures (larvae). Georeferencing of point coordinates was effectuated in decimal degrees of latitude and longitude in WGS84. World country map was based on simplified world country polygons by the National Aeronautics and Space Administration (NASA).

**Figure 2 insects-14-00049-f002:**
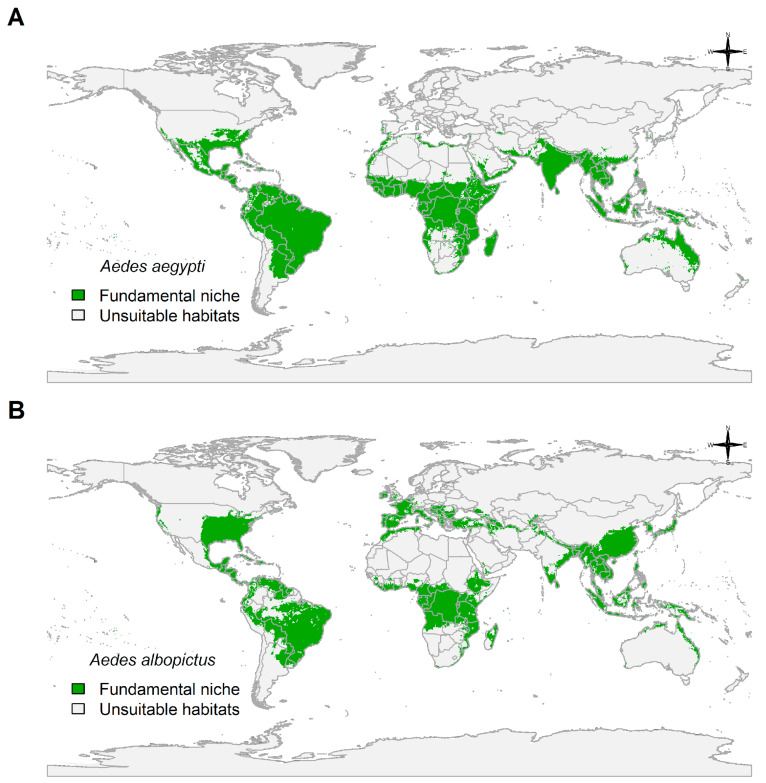
Spatial prediction of fitted boosted regression tree models: (**A**) *Aedes aegypti* (model evaluation: AUC = 0.9946303; threshold of maximum sensitivity + specificity = 0.8452035); and (**B**) *Aedes albopictus* (model evaluation: AUC = 0.9996303; threshold of maximum sensitivity + specificity = 0.9944801). Fundamental niche = predicted probability of presence > threshold; unsuitable habitats = otherwise. World country map was based on simplified world country polygons by NASA.

**Figure 3 insects-14-00049-f003:**
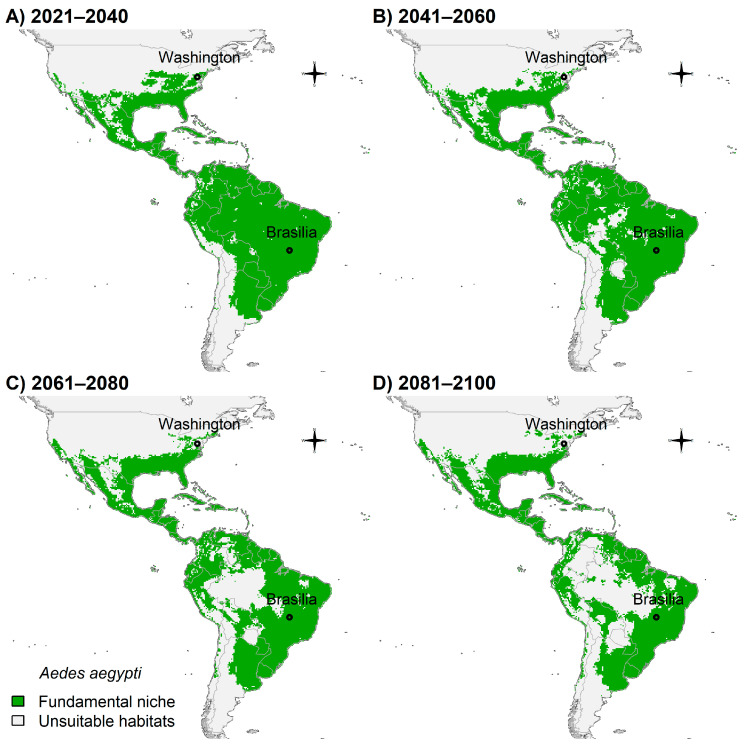
Potential future distribution of *Ae. aegypti* in Americas: (**A**) 2021–2040; (**B**) 2041–2060; (**C**) 2061–2080; and (**D**) 2081–2100. Future climate projections based on the global climate model GFDL-ESM4.1 by NOAA in the shared socio-economic pathway SSP3-7.0 of region rivalry. Locations of capitals of countries were shown for reference.

**Figure 4 insects-14-00049-f004:**
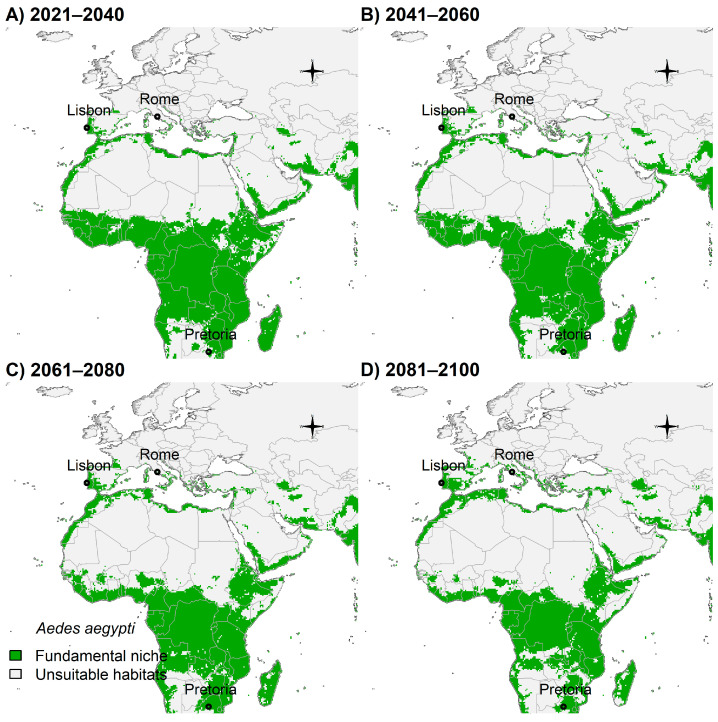
Potential future distribution of *Ae. aegypti* in Europe and Africa: (**A**) 2021–2040; (**B**) 2041–2060; (**C**) 2061–2080; and (**D**) 2081–2100. Future climate projections based on the global climate model GFDL-ESM4.1 by NOAA in the shared socio-economic pathway SSP3-7.0 of region rivalry. Locations of capitals of countries were shown for reference.

**Figure 5 insects-14-00049-f005:**
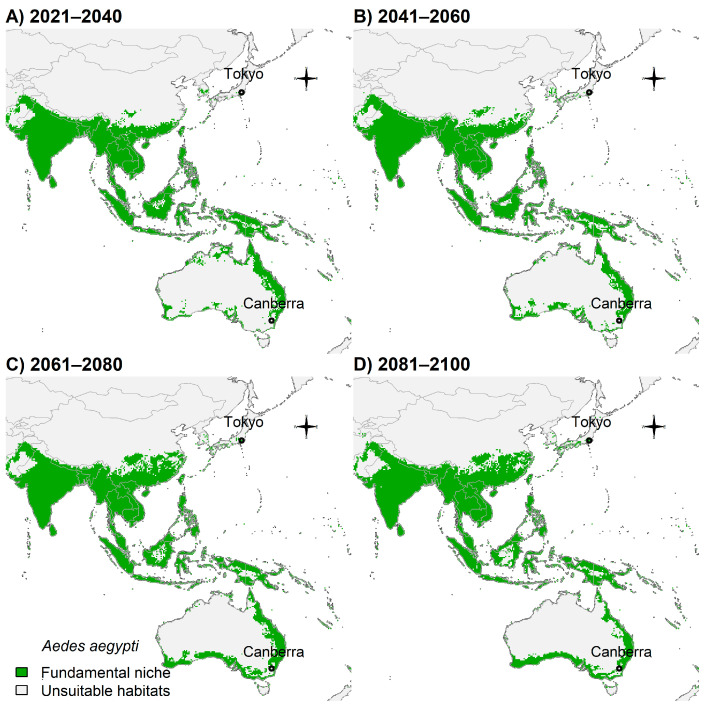
Potential future distribution of *Ae. aegypti* in Asia and Oceania: (**A**) 2021–2040; (**B**) 2041–2060; (**C**) 2061–2080; and (**D**) 2081–2100. Future climate projections based on the global climate model GFDL-ESM4.1 by NOAA in the shared socio-economic pathway SSP3-7.0 of region rivalry. Locations of capitals of countries were shown for reference.

**Figure 6 insects-14-00049-f006:**
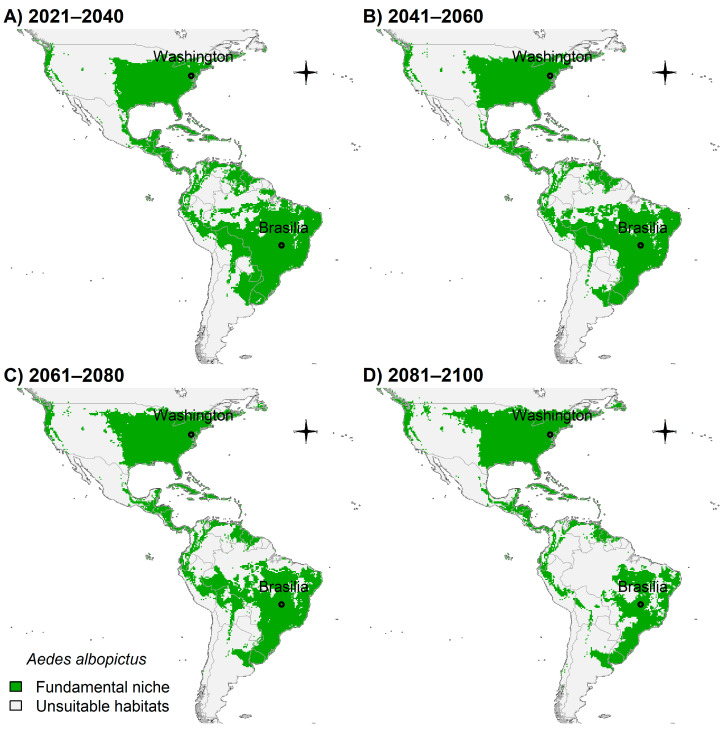
Potential future distribution of *Ae. albopictus* in Americas: (**A**) 2021–2040; (**B**) 2041–2060; (**C**) 2061–2080; and (**D**) 2081–2100. Future climate projections based on the global climate model GFDL-ESM4.1 by NOAA in the shared socio-economic pathway SSP3-7.0 of region rivalry. Locations of capitals of countries were shown for reference.

**Figure 7 insects-14-00049-f007:**
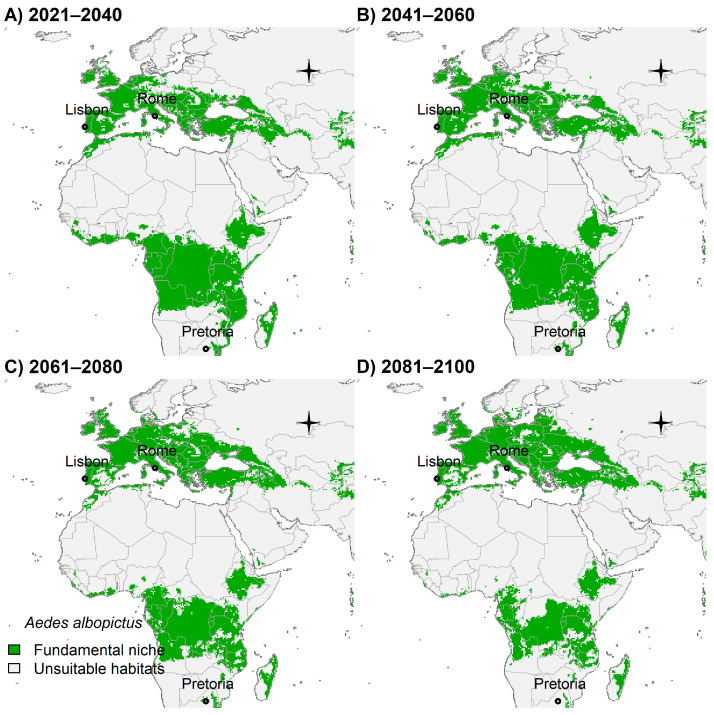
Potential future distribution of *Ae. albopictus* in Europe and Africa: (**A**) 2021–2040; (**B**) 2041–2060; (**C**) 2061–2080; and (**D**) 2081–2100. Future climate projections based on the global climate model GFDL-ESM4.1 by NOAA in the shared socio-economic pathway SSP3-7.0 of region rivalry. Locations of capitals of countries were shown for reference.

**Figure 8 insects-14-00049-f008:**
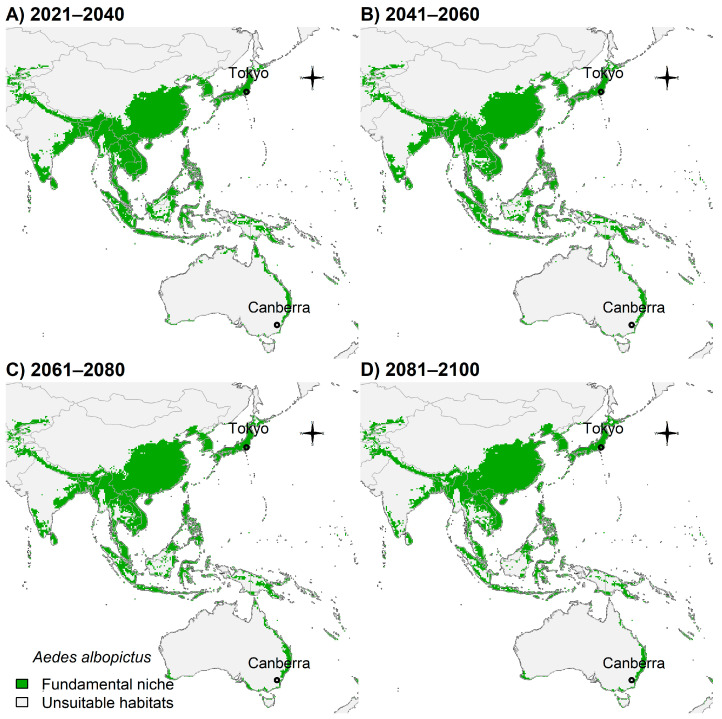
Potential future distribution of *Ae. albopictus* in Asia and Oceania: (**A**) 2021–2040; (**B**) 2041–2060; (**C**) 2061–2080; and (**D**) 2081–2100. Future climate projections based on the global climate model GFDL-ESM4.1 by NOAA in the shared socio-economic pathway SSP3-7.0 of region rivalry. Locations of capitals of countries were shown for reference.

## Data Availability

The data used for analysis can be available under reasonable request.

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
