# Peer review of "Global Distribution of Aedes aegypti and Aedes albopictus in a Climate Change Scenario of Regional Rivalry"

_insects, 2023, doi:10.3390/insects14010049_

Round 1

Reviewer 1 Report

1. This manuscript is valuable for reference, but the description of the prediction model is not clear enough. R software package or machine learning or deep machine learning method is used for prediction. Not innovative enough.

2.Although at 239-240:Evaluation of this model showed that its AUC was approximately and its threshold that maximises sensitivity + specificity was 0.995。The data given by the verification method is simple and has low credibility.It is recommended to display the validation data graph.

3.Further verify data and forecast results.

Author Response

Reviewer 1

1. This manuscript is valuable for reference, but the description of the prediction model is not clear enough. R software package or machine learning or deep machine learning method is used for prediction. Not innovative enough.

Response: We thank the reviewer. The prediction model and outcomes are better described in the revised version (Lines 99–114 in Introduction, Lines 159–165 in Methods, new Figures 3–8 in Results, Lines 369–392 in Discussion, Supplementary Material Figure S1–S10). Novelty can be attributed to the interpretation on the differences between Southern and Northern Hemispheres in relation to the expansion of these species’ potential distributions, as consequences from climate change may be different depending on the geographic locations. Also, it's important to recognize that mitigation of greenhouse gas emissions is not in the ‘priority agenda’ of the today’s leaders. Showing outcomes for Ae. aegypti and Ae. albopictus from a scenario of regional rivalry has some novelty as well (Lines 1–2, new title).

2. Although at 239-240:Evaluation of this model showed that its AUC was approximately and its threshold that maximizes sensitivity + specificity was 0.995. The data given by the verification method is simple and has low credibility. It is recommended to display the validation data graph.

Response: In the Figure S1 and Figure S4 (Supplementary Material) we show ROC-curve plots and density plots showing the prediction values of presences and pseudo-absences in the model for Ae. aegypti and Ae. albopictus. ROC-curve plots show that both sensitivity and specificity are high, as well as the AUC value. In the density plots it is shown that presence data are predicted correctly, and pseudo-absence prediction is mostly correct. These corrections increased credibility of the model outcomes.

3.Further verify data and forecast results.

Response: We verified data and forecast results as requested. They are correct. However, we revised our interpretation on the forecast results. Thus, we modified the title (Lines 1–2), Abstract (Lines 33–35), and Discussion (369–392), to better reflect our data and forecast. In addition, we added forecast results as a quantitative variable in Supplementary Material (Figures S7-S10).

Reviewer 2 Report

There are a few minor editing changes like making sure that all species names are not capitalized.  Additionally in the abstract please correct the sentence it should be we first built... not we firstly build.  Otherwise the paper is very well written and the findings can be used for areas of the world that may see an expansion of these important vector species.  

Author Response

Reviewer 2

There are a few minor editing changes like making sure that all species names are not capitalized. Additionally in the abstract please correct the sentence it should be we first built... not we firstly build. Otherwise, the paper is very well written, and the findings can be used for areas of the world that may see an expansion of these important vector species.

Response: We thank the reviewer. We have corrected species names and other typos or grammar mistakes, including the ones indicated. Additionally, we improved clarity in other parts of the text, including in the title (Lines 1–2) and in the Results (Figures 3–8).

Reviewer 3 Report

General comments:

In this study, the authors collected worldwide available data on Ae. albopictus and Ae. aegypti distribution data and created a model driven by temperature and precipitation to explain the distribution of those species. In a further step, they used this model to predict future species distribution for the next 100 years.

While they show interesting results, which could be useful for predicting future disease risk, the authors do not explain why individual specifications were chosen for the model (see specific comments below). The authors state that they are comparing different scenarios (title, line 97), however only one was chosen for this study (line 144). There is no explanation why this one was selected and how the selection of this specific scenario influences the results. What is missing the most is that the authors do not discuss their results in relation to previous studies on this topic.

Specific comments:

change title: The authors mix effects of changes in vector distribution, which is a result of their analysis (disease risk), with model input (desertification as a result of climatic change)

line 53-56: please explain why the desiccation resistance helps the spreading of those species

line 68: replace “domesticated” as this term assumes an intentional action from humans to change animals (or plants)  

line 72: what do “these conditions” refer to?

line 79: this is rather imprecise – min/max temperature? summer/winter?

line 81: use italic for “Ae. aegypti”

line 97: “… present possible scenarios.” (plural!) However, only one scenario is modelled.

line 110: please state when the search was conducted as the resulting number of data points will depend on this

line 124: How was occurrence data that was close to each other handled? E.g. if a study was conducted in a small area, many data points would be available for this area, while no data may exist from the surrounding areas. How does this heterogenity in the data collection bias the model results?

line 131: please state the final number of records for the two species

line 141 and 142: remove underline from °- symbol

line140-143: Please elaborate why those variables were chosen. How do they effect the mosquitoes?

line 144: Why was this scenario chosen?

line 163: change “absence” to “Pseudo-absence”

line 163-165: How was the exact data on occurrence probability for a specific location extracted for a specific location? Were the data from Kraemer et al made available by the authors of Kraemer’s study? The mentioned map shows continuous data, thus getting exact data by just looking at the colour at a location seems very imprecise.

I am not an expert in spatial models, but as far as I know pseudo-absences are usually generated from the own dataset. Why were the pseudo-absences in this study based on another study? Please provide an explanation and references.

line 187: change Albopictus and Aegypti to lower case letters

line 219 and 221: please provide not only the change, but  “from – to” numbers as well (it makes a big difference if precipitation is reduced from 169 to zero or from 1169 to 1000)

line 232-247: I would think that the influence of minimum or maximum temperature strongly depends on the months. E.g. that the minimum temperature in winter is much more important than in the summer months. Please provide results that are more detailed.

Figure 3+4: The sub-figures are rather small, please enlarge them. The legend on the other hand is tiny. As it is the same for all sub-figures I would suggest to just show the legend once in a much bigger front for all panels.

line 327: I would suggest to refer to more recent developments as well (autochthonous dengue chases in France 2022)

line 332: I would recommend to exchange the reference to a more credible source. “MasterCard Worldwide Insights” does not appear to be a scientific journal.  

line 332-333: This seems rather far fetched. If this is the case, please provide references.

line 349: explain why

line 397-382: Do that! (It seems that this is the comment of another reviewer, which has been copied into the discussion instead of being implemented.)

line 516-518: edit reference 46

Author Response

Reviewer 3

General comments:

In this study, the authors collected worldwide available data on Ae. albopictus and Ae. aegypti distribution data and created a model driven by temperature and precipitation to explain the distribution of those species. In a further step, they used this model to predict future species distribution for the next 100 years.

While they show interesting results, which could be useful for predicting future disease risk, the authors do not explain why individual specifications were chosen for the model (see specific comments below). The authors state that they are comparing different scenarios (title, line 97), however only one was chosen for this study (line 144). There is no explanation why this one was selected and how the selection of this specific scenario influences the results. What is missing the most is that the authors do not discuss their results in relation to previous studies on this topic.

Response: Thank you for the reviewer. Yes, only one scenario was selected, thank you for your observation. We have corrected this mistake throughout the text. In addition, we added specific discussion of our results in the first paragraphs of the Discussion section.

Specific comments:

change title: The authors mix effects of changes in vector distribution, which is a result of their analysis (disease risk), with model input (desertification as a result of climatic change)

Response: Thank you! Yes, totally agreed. The title was revised accordingly (Lines 1–2).

line 53-56: please explain why the desiccation resistance helps the spreading of those species

Response: Now this link is well explained with proper citation (Lines 53–59).

line 68: replace “domesticated” as this term assumes an intentional action from humans to change animals (or plants) 

Response: we modified this term by “synanthropic adaptation”, which seems more appropriate (Line 70).

line 72: what do “these conditions” refer to?

Response: We specified it as “extreme events of above-average temperatures” and their outcomes as “Ae. aegypti populations to adapt to new habitats including underground oviposition and resting sites” (Lines 74–75).

line 79: this is rather imprecise – min/max temperature? summer/winter?

Response: Now it reads “winter and summer temperatures” (Line 82).

line 81: use italic for “Ae. aegypti”

Response: It was italicized (Lines 85).

line 97: “… present possible scenarios.” (plural!) However, only one scenario is modelled.

Response: We corrected this to reflect the scenario chosen. We added more information to provide more details on this scenario and why it was selected in Introduction (Lines 99–114).

line 110: please state when the search was conducted as the resulting number of data points will depend on this

Response: Now it is stated as January 12th, 2022 (Line 125).

line 124: How was occurrence data that was close to each other handled? E.g. if a study was conducted in a small area, many data points would be available for this area, while no data may exist from the surrounding areas. How does this heterogeneity in the data collection bias the model results?

Response: We excluded all the duplicated and non-land records. By doing that, we removed errors and biased data of the original dataset. We ended with 9,735 presence records of Ae. aegypti and 13,093 presence records of Ae. albopictus to be modeled. Although these remaining datasets have some heterogeneity, as mentioned, boosted regression trees can deal with heterogeneity in the data for species distribution modelling.

line 131: please state the final number of records for the two species

Response: This was stated as: “The final species occurrence data for Ae. aegypti (N=9,735) and Ae. albopictus (N=13,093) were free of duplicated and non-land georeferenced latitude and longitude” (Lines 145–147).

line 141 and 142: remove underline from °- symbol

Response: This was corrected here (Line 52) and throughout the text.

line 140-143: Please elaborate why those variables were chosen. How do they effect the mosquitoes?

Response: It now reads: “These variables were selected because of their strong link with the ecological niche of both species, thereby having a combined relative contribution as high as 77.6% and 74.3% among all covariates on Ae. aegypti and Ae. albopictus global distribution, respectively [1]. They were employed to build a fundamental niche model and then estimate future potential distributions.” (Lines 154–158)

line 144: Why was this scenario chosen?

Response: We included a new paragraph to justify the selection of this paragraph (Lines 159–165).

line 163: change “absence” to “Pseudo-absence”

Response: Absence was replaced by “pseudo-absence” here (Line 180) and throughout the text.

line 163-165: How was the exact data on occurrence probability for a specific location extracted for a specific location? Were the data from Kraemer et al made available by the authors of Kraemer’s study? The mentioned map shows continuous data, thus getting exact data by just looking at the colour at a location seems very imprecise.

I am not an expert in spatial models, but as far as I know pseudo-absences are usually generated from the own dataset. Why were the pseudo-absences in this study based on another study? Please provide an explanation and references.

Response: We selected random points in where Kraemer et al. predicted the lowest probability of presence (i.e., blue color). We used the maps and data provided in the published article. We included more information on this and added two references: “This approach is used for generating the non-presence class for generalized boosted regression models [32]. It means unsuitable places where species absences are more likely, aka pseudo-absence [33].” (Lines 182–184)

  1. Laporta, G.Z.; Linton, Y.-M.; Wilkerson, R.C.; Bergo, E.S.; Nagaki, S.S.; Sant’Ana, D.C.; Sallum, M.A.M. Malaria Vectors in South America: Current and Future Scenarios. Parasit Vectors 2015, 8, 426, doi:10.1186/s13071-015-1038-4. (Lines 596–598)
  2. Phillips, S.J.; Dudík, M.; Elith, J.; Graham, C.H.; Lehmann, A.; Leathwick, J.; Ferrier, S. Sample Selection Bias and Presence-Only Distribution Models: Implications for Background and Pseudo-Absence Data. Ecol Appl 2009, 19, 181–197, doi:10.1890/07-2153.1. (Lines 599–601)

line 187: change Albopictus and Aegypti to lower case letters

Response: These are lower case in the revied version. (Line 207)

line 219 and 221: please provide not only the change, but “from – to” numbers as well (it makes a big difference if precipitation is reduced from 169 to zero or from 1169 to 1000)

Response: The range (from - to) is now provided: “In Washington DC, the capital of the United States, annual average temperature and annual total precipitation increased from 1,100 mm and 13.2 °C in 1970–2000 to 1,269 mm and 17.1 °C in 2081–2100. In Brasilia, the capital of Brazil, annual average temperature al-so increased by 3.9 °C (range: 20.3–24.2 °C), but annual total precipitation decreased by 207 mm (range: 1539–1332 mm) in 2081–2100 vs. 1970–2000.” (Lines 236–241)

line 232-247: I would think that the influence of minimum or maximum temperature strongly depends on the months. E.g. that the minimum temperature in winter is much more important than in the summer months. Please provide results that are more detailed.

Response: For Ae. aegypti, we added: “The most important predictor was maximum temperature in October (relative contribution=15.4%). Relative contribution and response curves per top 15 predictors are shown in Figure S2 (Supplementary Material). A more precise assessment at fitted values in relation to the most important predictor in the model (maximum temperature in October) is in Figure S3 (Supplementary Material).” (Lines 255–259)

For Ae. albopictus, it was added: “The top 3 predictors were: minimum temperature in October (relative contribution=22.6%), precipitation in April (20%), and minimum temperature in September (17.8%) (Figure S5 and Figure S6–Supplementary Material).” (Lines 271–273)

Figure 3+4: The sub-figures are rather small, please enlarge them. The legend on the other hand is tiny. As it is the same for all sub-figures, I would suggest to just show the legend once in a much bigger front for all panels.

Response: Yes! We corrected this. We now have Figures 3–8 that show results in finer scale maps. (Lines 309–361)

line 327: I would suggest to refer to more recent developments as well (autochthonous dengue chases in France 2022)

Response: Absolutely. We refer to these cases and to this reference: “This species is contributing to an increased risk of vector-borne disease transmission across mainland France [43]. A total of 65 autochthonous dengue cases (serotypes 1 and 3) were reported in France, 2022, and the dengue virus identified in these humans was probably vectorized by local Ae. albopictus [43]” (Lines 414–417)

line 332: I would recommend to exchange the reference to a more credible source. “MasterCard Worldwide Insights” does not appear to be a scientific journal. 

Response: We understand this is not a scientific journal, but it is a good proxy for supporting our statement, so we maintained this approach with corrections in the published date and url of this reference: “As many European countries are among the most visited destinations in the world [44]”

  1. Robino, D.M. Global Destination Cities Index 2019. MasterCard. https://www.mastercard.com /news/media/wexffu4b/gdci-global-report-final-1.pdf. (Lines 628–629)

line 332-333: This seems rather far fetched. If this is the case, please provide references.

Response: We changed the meaning of this sentence, and it now reads: “As many European countries are among the most visited destinations in the world [44], further invasion and transmission of serotypes of dengue or other arboviruses is likely.” (Lines 417–419)

line 349: explain why

Response: We added herein: “The life cycle of these mosquitoes depends on the availability of habitats for eggs, larvae, and pupae [16]. The development of these immatures is only achieved when an optimum range of precipitation occurs [16]. Adult males and females depend on optimum ranges of temperature to fly into the air; otherwise, they will become inactive and/or die [16].” (Lines 436–438)

line 397-382: Do that! (It seems that this is the comment of another reviewer, which has been copied into the discussion instead of being implemented.)

Response: Oops. We excluded this sentence, but we better discussed our results considering the literature in the first sentences of the Discussion section. (Lines 369–393)

line 516-518: edit reference 46

Response: Reference 46 was edited accordingly and not reads:

  1. Grillet, M.E.; Hernández-Villena, J.V.; Llewellyn, M.S.; Paniz-Mondolfi, A.E.; Tami, A.; Vincenti-Gonzalez, M.F.; Marquez, M.; Mogollon-Mendoza, A.C.; Hernandez-Pereira, C.E.; Plaza-Morr, J.D.; et al. Venezuela’s Humanitarian Crisis, Resurgence of Vector-Borne Diseases, and Implications for Spillover in the Region. Lancet Infect Dis 2019, 19, e149–e161, doi:10.1016/S1473-3099(18)30757-6. (Lines 632–635)

Reviewer 4 Report

The title is not necessary fitting the content of the work.
The environmental variables analyzed and used to predict future distributions are temperature and precipitation, so Forest land cover type is not analyzed directly, in the methodology neither in the results

Methodology:
Is clearly explained and the model and validation are in accordance to recent works in this area.
The authors must have care about using a big dataset of mosquito pretense, even they do some cleaning of the information downloaded from Vector map. Some presence points outside the Known distribution of both species should be verified. For example: some points at north France or at south of Argentina for Ae aegypti. These presence points should have a lot of influence in the actual and future prediction because there are already placed where physical variables have conditional/ or critical values for the presence of these species.

Results: the optimum range  for temperatures in Ae si quite broad, But this result could be influenced by the non homogeneous distribution of precense data, and by the pseudo absence data. (As the authors mention in the discussion).
The maps are interesting but one or two if these figures could have better resolution.  A map using probability of pretense (more than a binary map of pretenda/abcense) could be more informative. Also provide a map showing the error or variance of the prediction  would make the work much rigorous (Im not sure if it is possible for boosted regression tree?

Some map of temperature variables in futures periods, could bring some aditional  information to the future Ae and A albopictus distribution.

Discussion: Is very interesting the loose of habitats suitability for Aedes aegypti in Amazonian region. But the authors shoud focus part of this section in the changes of climatic variables used to model the future Niche of the Ae (i. e. Temperature and precipitation). Desertification, and habitat/forest change  will clearly affect biodiversity and also will play an important in many species distribution range. But, at least Aedes aegypti is clearly adapted to human habitat now a days, so this biodiversity/forest/ Aedes aegypti habitat is not so linear for me.

Author Response

Reviewer 4

The title is not necessary fitting the content of the work.

The environmental variables analyzed and used to predict future distributions are temperature and precipitation, so Forest land cover type is not analyzed directly, in the methodology neither in the results

Response: Agreed. The title was changed to better reflect our work. (Lines 2–3). In addition, our interpretation about the results in the Abstract and Discussion was corrected accordingly.

Methodology:

Is clearly explained and the model and validation are in accordance to recent works in this area.

The authors must have care about using a big dataset of mosquito pretense, even they do some cleaning of the information downloaded from Vector map. Some presence points outside the Known distribution of both species should be verified. For example: some points at north France or at south of Argentina for Ae aegypti. These presence points should have a lot of influence in the actual and future prediction because there are already placed where physical variables have conditional/ or critical values for the presence of these species.

Response: Data in VectorMap are curated by entomologists and museum specialists.

As for the occurrence of Ae. aegypti in France, it was provided by Institute of Research for Development in Montpellier, France. This institute has a preserved specimen of this species which was collected in France in the coordinates 48.9N and 2.5E. This record can be accessed via GBIF Dataset http://www.gbif.org/dataset/4b3541f2-92e0-4cc6-bb10-a2a2ffe61c60

As for the occurrence of this species in the south of Argentina, this record was provided by the Walter Reed Biosystematics Unit (http://www.wrbu.org/) and it was used in the published model by Kraemer et al. (Kraemer, Moritz U.G. et al. 2015. The global compendium of Aedes aegypti and Ae. albopictus occurrence. Scientific Data, 2:150035 DOI: 10.1038/sdata.2015.35.)

Results: the optimum range for temperatures in Ae is quite broad, but this result could be influenced by the non-homogeneous distribution of presence data, and by the pseudo absence data. (As the authors mention in the discussion).

Response: it is broad because these species really occur in most of the tropical and sub-tropical areas of the world. Additionally, we are modelling their potential fundamental niche, which can be interpreted as places suitable for the species when they reach at them.

The maps are interesting but one or two if these figures could have better resolution.  A map using probability of presence (more than a binary map of presence/absence) could be more informative. Also provide a map showing the error or variance of the prediction would make the work much rigorous (I’m not sure if it is possible for boosted regression tree?

Response: Yes. Correct. We added maps of continuous probability of presence for both species in the Supplementary Material (Figure S7–Figure S10) and Lines 363–365.

We added information on the performance of the prediction in the Supplementary Material (Figure S1, Figure S4). In fact, boosted regression trees showed high performance of prediction with our data. In Figure S1 and Figure S4, ROC-curve plots and density plots are shown. In density plots it can be observed that presence was correctly predicted and most of pseudo-absence was predicted correctly. This shows that the results and outcomes have credibility.

Some map of temperature variables in futures periods, could bring some additional information to the future Ae and A albopictus distribution.

Response: Yes. We added more information on the relationship between mosquito data and environmental variables in Supplementary Material (Figure S2, Figure S3, Figure S5, and Figure S6).

Discussion: Is very interesting the loose of habitats suitability for Aedes aegypti in Amazonian region. But the authors should focus part of this section in the changes of climatic variables used to model the future Niche of the Ae (i. e. Temperature and precipitation). Desertification, and habitat/forest change will clearly affect biodiversity and will play an important in many species distribution range. But at least Aedes aegypti is clearly adapted to human habitat now a days, so this biodiversity/forest/ Aedes aegypti habitat is not so linear for me.

Response: Yes. We changed the beginning of the Discussion section and added two new paragraphs to better reflect this comment (Lines 369–393). The Abstract and the Title were changed as well. In the future the Amazon will be drier, and it will be less suitable for Aedes (Stegomyia) spp. We removed the part referring to biodiversity, as this was somewhat confusing.

Round 2

Reviewer 1 Report

/